# Sampling Low Air Pollution Concentrations at a Neighborhood Scale in a Desert U.S. Metropolis with Volatile Weather Patterns

**DOI:** 10.3390/ijerph19063173

**Published:** 2022-03-08

**Authors:** Nathan Lothrop, Nicolas Lopez-Galvez, Robert A. Canales, Mary Kay O’Rourke, Stefano Guerra, Paloma Beamer

**Affiliations:** 1Mel and Enid Zuckerman College of Public Health, University of Arizona, Tucson, AZ 85721, USA; nilopez@sdsu.edu (N.L.-G.); mkor@arizona.edu (M.K.O.); stefano@arizona.edu (S.G.); 2School of Public Health, San Diego State University, San Diego, CA 92182, USA; pbeamer@email.arizona.edu; 3Program in Applied Mathematics, University of Arizona, Tucson, AZ 85721, USA; robert.canales@gmail.com; 4Asthma and Airway Disease Research Center, College of Medicine, University of Arizona, Tucson, AZ 85721, USA

**Keywords:** air pollution monitoring, oxides of nitrogen, particulate matter, climate change

## Abstract

Background: Neighborhood-scale air pollution sampling methods have been used in a range of settings but not in low air pollution airsheds with extreme weather events such as volatile precipitation patterns and extreme summer heat and aridity—all of which will become increasingly common with climate change. The desert U.S. metropolis of Tucson, AZ, has historically low air pollution and a climate marked by volatile weather, presenting a unique opportunity. Methods: We adapted neighborhood-scale air pollution sampling methods to measure ambient NO_2_, NO_x_, and PM_2.5_ and PM_10_ in Tucson, AZ. Results: The air pollution concentrations in this location were well below regulatory guidelines and those of other locations using the same methods. While NO_2_ and NO_x_ were reliably measured, PM_2.5_ measurements were moderately correlated with those from a collocated reference monitor (r = 0.41, *p* = 0.13), potentially because of a combination of differences in inlet heights, oversampling of acutely high PM_2.5_ events, and/or pump operation beyond temperature specifications. Conclusion: As the climate changes, sampling methods should be reevaluated for accuracy and precision, especially those that do not operate continuously. This is even more critical for low-pollution airsheds, as studies on low air pollution concentrations will help determine how such ambient exposures relate to health outcomes.

## 1. Introduction

Exposure to ambient air pollutants has long been implicated in numerous health effects [1,2,3], especially respiratory outcomes; however many long-running birth cohorts lack exposure data, necessitating modeling with other air pollution measures [4,5]. Similar to other birth cohorts, the Tucson Children’s Respiratory Study has followed subjects since birth in 1980–1984 but has no air pollution exposures. To take advantage of this long-running health dataset, we sought to measure air pollution at a neighborhood scale to ultimately model exposures for health effect analyses.

The Tucson, Arizona, metropolitan area (Tucson), in the desert southwestern U.S. is home to 1 million people [6] and is defined by a car-centric, low-density, suburban morphology common in the western U.S. While the ambient concentrations of most air pollutants are historically low, Tucson is prone to high naturally occurring, windblown dust concentrations and a climate marked by volatile precipitation patterns, extreme aridity and heat, and dust storms [7,8,9]. With climate change, meteorology around the world will become more extreme [10,11,12,13], mimicking aspects of Tucson’s current climate. There are few studies assessing measurements in arid environments [14,15] and none, to our knowledge, in low-pollution airsheds. Low-pollution locations offer the chance to study threshold effects of pollutant concentrations below regulatory levels—a potential preview of health effects if air pollution is reduced. This presents an opportunity to examine measurement methods at a neighborhood scale. Past studies have used a variety of sampling methods to measure pollutants with intra-city spatial precision, ranging from passive, low-cost samplers (e.g., Ogawa badge samplers) to active monitors (e.g., chemiluminescence monitors) [16,17,18,19,20,21,22,23]. Given the wide range of air sampling approaches, we emulated well-documented protocols implemented in a variety of settings: the European Study of Cohorts for Air Pollution Effects (ESCAPE).

ESCAPE methods for all aspects of monitoring, including location designation, setup, sample analysis, and later model development, have been used in 36 locations among 16 countries throughout Europe, ranging from bucolic to densely urban, from Stockholm in Sweden to the Grecian isle of Crete [24,25]. More recently, these methods were implemented in Taipei, Taiwan [26], in the coastal cities of Perth [27,28] and Brisbane [29] in Australia, and in Durban, South Africa [30]. However, these methods have not been adapted in locations with low air pollution concentrations and more extreme meteorological patterns. In this paper, we illustrate how we adopted and adapted ESCAPE air pollution measurement methods for ambient nitrogen dioxide (NO_2_), oxides of nitrogen (NO_x_), and particulate matter < 2.5 µm (PM_2.5_) and <10 µm (PM_10_) in diameter, and then assess and compare our findings to ESCAPE and other relevant studies.

## 2. Materials and Methods

The Tucson, AZ, metropolitan region (Tucson), with an area of 5275 km^2^, is located in the eastern half of Pima County in southern Arizona, approximately 100 km north of the U.S.–Mexico border (Figure 1). It is defined by the City of Tucson, smaller surrounding jurisdictions, and unincorporated residential development. While the region has a mean density of 190 persons/km^2^, the City of Tucson (pop. = 535,677) has 809 persons/km^2^ [6]. Residential development largely comprises detached, single-family dwellings, with little mixed land use. NO_2_, NO_x_, and PM_2.5_ are byproducts of combustion from industry and vehicle travel, as well as from reactions in the atmosphere. While Tucson lacks heavy industry and large point sources of pollution, Tucson has an extensive street network for personal vehicle travel as the primary mode of transportation, which generates area source pollution, much like other urban areas in the western U.S. Nevertheless, these pollutants have been perpetually below relevant ambient air quality standards [7].

Increased concentrations of PM_10_ in the study area often come from a combination of high winds, limited precipitation, and increased aridity [7]. Together, these factors result in less PM_10_ and PM_2.5_ being scoured from the air, less hygroscopic particulate growth [31];,and less dust-arresting vegetation cover [32]. In the early summer, daytime high temperatures regularly reach 43 °C, while snowfall can be expected at least once over winter [8]. While Tucson receives an average total of 30 cm of precipitation, nearly all rainfall comes in the form of violent, but short-lived, summer monsoons or steady, gentler winter rains [8]. In the future, Tucson’s annual precipitation is expected to be reduced even further compared to that in other parts the western U.S. [33]. In addition, dust storms or “*haboobs*” are more frequent, particularly during the summer monsoon season [9]. 

Following methods from ESCAPE [24,25,34], we aimed to sample pollutants at 40 sites within the Tucson, AZ, metropolitan area to ensure sufficient geographic representation while balancing the feasibility of sampler set up and material costs. Sites were divided into three distinct types, defined by their expected sources of air pollution: regional background, urban background, and street (see Appendix A for details). Sites were recruited to ensure that half of the locations of each site type measured NO_2_ and NO_x_ and the other half, NO_2_, NO_x_, PM_2.5_, and PM_10_. In Tucson, there are few building canyons. As a result, unlike ESCAPE, in which sites could include balconies, our study utilized only ground-level locations. 

Sites were selected from residential areas where two birth cohorts lived by overlaying cohort addresses with vehicle traffic maps using ArcGIS ArcMap 10.4 (ESRI, Redlands, CA, USA). We iteratively recruited via email through multiple listservs at the Mel and Enid Zuckerman College of Public Health and the Asthma and Airways Disease Research Center at the University of Arizona, as well as with word-of-mouth recruitment among existing participants and study personnel. As recruitment progressed, later emails solicited residents in specific areas where site representation was lacking. As a result, sampling sites were private residences. 

Similar to ESCAPE, we sampled during 14-day contiguous periods, in which we concurrently measured NO_2_ and NO_x_ at five sites and NO_2_, NO_x_, PM_2.5_, and PM_10_ at another five sites [24,34]. We scheduled sampling at the participant’s convenience, and so the proportion of site types matched the whole study and that sampling setup types were geographically representative. Each site was sampled for three periods, with each period in a different season. In ESCAPE, sampling seasons were divided into four months of “winter,” four months of “summer,” and four months of “intermediate” season (i.e., spring and fall) [24,34]. However, Tucson has historically had five seasons: a wet and cool winter (December–February), a windy and temperate spring (March–April), a dry and hot pre-monsoon (May–June), a wet and hot monsoon (July–August), and a windy and temperate fall (September–November) [8]. 

To adapt, we defined summer as May through August, winter as November through February, and the remaining months as the intermediate season. Sampling in summer was distributed evenly among pre-monsoon and monsoon months. During each sampling period, another sampler setup measuring NO_2_, NO_x_, PM_2.5_, and PM_10_ was collocated at a U.S. Environmental Protection Agency (USEPA) National Core regulatory monitoring station in the center of the city, which is designed to accurately measure low concentrations of NO_2_, NO_x_, and PM_2.5_ using trace-level instrumentation [7]. In doing so, we were able to assess study measurement method accuracy against reference methods and calculate temporally corrected annual mean concentrations for each site for comparison to other ESCAPE-based studies (see Appendix A for details). Sampling began in September 2015 and concluded one year later.

Following ESCAPE methods, we measured NO_2_ and NO_x_ using Ogawa Badge Samplers (Ogawa & Company, USA, Inc., Pompano Beach, FL, USA) at a height of 2 m, which were analyzed at the University of Arizona (see Supplemental Materials for details). Particulate matter was sampled using two 10 L/min–rated Harvard impactors (Air Diagnostics and Engineering, Inc., Naples, ME, USA): one for PM_2.5_ and one for PM_10_. Unlike ESCAPE, which used a customized single pump to run both impactors, we used separate SKC Leland Legacy pumps (SKC Inc., Eighty Four, PA, USA), which were calibrated immediately before and after sampling at the site using a Mesa Labs DryCal Defender 520 (Mesa Laboratories, Inc., Butler, NJ, USA). The pumps were programmed to sample for a contiguous 15 min period out of every two hours [34]. 

Impactors were suspended 2 m above the ground and 1 m apart, such that the inlets faced downward to prevent rain from soaking the filters. To adapt to Tucson’s extreme summer heat (daytime high temperatures regularly reach 43 °C) and avoid pump overheating and damage (rated to 45 °C), we ventilated pump cases with ten 2.5-cm-diameter holes, equally spaced along the perimeter of the vertical sides of the case, along the top. A constantly running PC fan was placed inside the container with the pumps to ensure continuous air flow. This container was surrounded by a frame of PVC pipe, and reflective Mylar-foil-covered bubble-pocket insulation was suspended between the pipes to prevent direct sun exposure while also allowing air to circulate around the pump container (Figure 2). In addition, to avoid the need of a PC to program pumps from overheating during summer months, samplers could be set up or taken down only from 5:30 to 10:00 a.m.

To collect PM, we used Whatman 37-mm-wide Teflon filters with a 0.2 micron pore size and support ring (Whatman, Inc., Maidstone, UK), because the ESCAPE-suggested Andersen filters were unavailable. Weighing filters in Tucson, AZ, where the relative humidity is <20% most of the year, required a static de-ionizer to remove static charge prior to each time the filter was weighed. All filters were weighed three times using a Mettler Toledo XP2U Ultra Micro Balance accurate to 0.1 µg (Mettler Toledo, Columbus, OH, USA), and then, the weights were averaged, provided the smallest and largest weights were within 10% of each other. Before weighing, all filters were conditioned for ≥48 h in a climate-controlled chamber. In addition, two lab blank filters were weighed, and their mean difference was used to correct the filter mass post-sampling for changes due to humidity. Unlike the ESCAPE approach, which loaded filters into impactors in the field, we loaded impactors in the laboratory and sealed them in plastic bags until set up at the site to reduce incidental contamination of filters from wind-blow dust when loading the impactors and to speed up sampler installation at the site under extreme heat conditions. Nearly all samples met ESCAPE quality assurance and control metrics (see Appendix A for details).

## 3. Results

Meteorological measures were taken at the Tucson International Airport, 16 km south of the collocated sampler at the reference monitor site (Table 1 and Figure 3). Winter months were characterized by small amounts of rainfall and cool temperatures, as well as increased wind speeds in November and late January. Sampling periods in spring were relatively dry, with a low dew point and high winds. There was no recorded precipitation during sampling periods in February, April, and May. In the pre-monsoon, temperatures and aridity peaked but were tempered by early monsoon rains that began in June. July was the wettest month, with substantial precipitation variability by the hour. Unexpectedly, August had little rain but was humid and had the second highest mean wind speeds. Fall sampling periods saw the end of the monsoon rains and high summer temperatures.

At the sampling site collocated with a reference monitor, the study monitor geometric mean (standard deviation) concentrations were NO_2_ = 3.91 (1.86) ppb, NO_x_ = 7.24 (2.14) ppb, and PM_2.5_ = 4.79 (1.36) μg/m^3^, and all concentrations were below the relevant USEPA National Ambient Air Quality Standards (NO_2_ = 53 ppb; PM_2.5_ = 12 μg/m^3^). Measurements from our monitor and the reference setup were moderately to highly correlated (Spearman correlations), with NO_2_ r = 0.80 (*p* = 0.003) and NO_x_ r = 0.88 (*p* = 0.002), while PM_2.5_ had an r = 0.41 (*p* = 0.13) (Figure 4). Correlations were nearly identical after including four additional measurement periods during November and December during which no other sites were sampled to avoid holidays as per ESCAPE protocols.

Compared to the reference monitor, the study setup tended to undermeasure NO_2_ and NO_x_ (mean percent differences of −45% and −15%, respectively), with the absolute percent difference decreasing as TAPS concentrations increased (NO_2_ r = 0.66, *p* = 0.02 and NO_x_ r = 0.58, *p* = 0.05). The pre-monsoon months of May and June had the greatest average percent differences: NO_2_ = −52% and NO_x_ = −34%. For PM_2.5_, the TAPS monitor overestimated on average by 37% but, with absolute percent difference, negatively correlated with reference monitor concentrations (r = −0.69, *p* = 0.02). TAPS-monitored PM_2.5_ concentrations during July through September were an average of 92% greater than the regulatory monitor’s, compared to 18% greater the rest of the time.

Throughout the study area, we measured NO_2_ and NO_x_ at 4 regional background, 23 urban background, and 12 street sites, and PM at 4 regional background, 11 urban background, and 4 street sites (Figure 3). Due to the study design, street sites tended to have limited secure space for the PM sampler setup, so these locations often accommodated only NO_2_ and NO_x_ samplers. NO_2_ and NO_x_ concentrations were significantly different between seasons, with the highest in winter and the lowest in summer (Wilcoxon signed-rank test, paired samples, Bonferroni correction, all *p* < 0.001), while PM concentrations were not (all *p* > 0.22) (Figure 5). During all seasons, street sites had higher NO_2_ and NO_x_ concentrations, followed by urban and then regional backgrounds, though this was not significant (Kruskal–Wallis test). PM_2.5_ concentrations exhibited this same relationship in winter and intermediate seasons, but this was only significant in winter (*p* = 0.05). PM_10_ also showed this trend only in winter, and it was not significant. In summer, there were no trends by site type for PM measures.

NO_2_ and NO_x_ measurements were highly correlated with one another (r = 0.94, *p* < 0.0001), with similar correlations among site types. With regard to the seasons, the correlation was the greatest in the intermediate season (r = 0.89, *p* < 0.0001), followed by that in winter (r = 0.84, *p* < 0.0001) and summer (r = 0.71, *p* < 0.0001). NO_2_/NO_x_ ratios were not significantly different by site type or season (Figure 6).

When comparing annual means that were temporally corrected and those that were not, the former were almost always higher (Table 2). These two annual means were highly correlated for NO_2_ (r = 0.94), NO_x_ (r = 0.95), and PM_10_ (r = 0.96) but less so for PM_2.5_ (r = 0.65). As shown previously, the NO_2_, NO_x_, and PM_2.5_ annual means at street sites were the highest, followed by those at urban and then regional backgrounds; however these differences were not significant. Meanwhile, PM_10_ had lower concentrations at urban background sites, but this was not significant. 

## 4. Discussion

In this study, we adopted and adapted ESCAPE methods previously used in Europe, coastal cities Australian and South Africa, and densely populated Taipei, Taiwan, to the suburban, low-pollution Tucson metropolitan area in the desert southwestern U.S. While ESCAPE methods with updates for varying seasonality and extreme heat were effective in Tucson’s climatological extremes, measurements of fine particulate matter using an ESCAPE monitor were poorly correlated with those from the collocated reference monitor. 

One reason may be that the reference monitor inlet was 4 m off the ground, which could result in the TAPS monitor collecting a greater number of larger particles nearer the ground, resulting in consistently higher PM_2.5_ concentrations. However, there was no relationship between the TAPS PM_2.5/10_ ratio and the reference monitor PM_2.5_ concentrations. Alternatively, sampling 15 min of every 2 h could have resulted in oversampling short-lived high PM_2.5_ sources (e.g., high-wind events or traffic emissions). This was performed as per ESCAPE protocols so that filters would not be overloaded over the 2-week sampling period.

In a past Tucson-based study, PM samplers were run continuously for a shorter sampling period using a Harvard impactor for lower flow rate (e.g., 4 L/min for 1 week) [35]. While PM_2.5_ was not reported, PM_10_ concentrations were higher than those in TAPS, indicating that this approach may not have overloaded filters. In our study, the absolute percent difference between the collocated monitors was negatively related to the reference monitor’s concentration, indicating the sampling method may lose accuracy in low-pollution airsheds, which are critical in demonstrating potential concentration thresholds in health effects. 

In comparison to other desert locations, albeit coastal ones, Tucson’s mean PM_2.5_ and PM_10_ concentrations are about an order of magnitude lower (42.0 and 161 µg/m^3^ in Chile and 41.6 and 138 µg/m^3^ in Kuwait, respectively [14,15]). In Kuwait, using a Harvard Impactor (Air Diagnostics and Engineering, Inc., Naples, ME, USA) modified for high-dust settings for 24 h samples, authors found that PM_2.5_ measures had better agreement with the collocated regulatory monitor compared to PM_10_, likely due to the greater spatial distribution of that size fraction [15]. Further, their impactor setup consistently under measured PM_2.5_, with the greatest differences during dust storms, as opposed to ours which over measured compared to the regulatory monitor. Unfortunately, we do not have regulatory PM_10_ concentrations at the site to compare. 

Interestingly, there were no dust storms or other abnormal or historic weather events, as defined by the NOAA Storm Events Database, during our study. However, the hottest months had the greatest PM_2.5_ percent difference, suggesting that the pumps may be less reliable when approaching their specified maximum operating temperature of 45 °C. While the maximum ambient temperature during sampling was 43 °C (May), temperatures inside cases could reach 45 °C if the ventilation fan failed. While we found no evidence of fan breakdown, it is possible that certain areas experienced power outages during periods of extreme heat. Unfortunately, there are no instrument diagnostics available for more information on whether temperature exceedances caused equipment issues. However, there was no linear trend between temperature and correlation. While pump diaphragms did need to be replaced periodically, this did not affect any of the reference monitor pumps.

It is notable that in Tucson and other southwestern U.S. cities, only 20–50% of PM variation can be explained by meteorological variables, and the remainder may come from unpredictable sources such as nearby vehicle traffic, forest fires, or transport from other locations [36]. While poor sampling agreement at the collocated site may be an isolated incident, it could systemically result in PM_2.5_ exposure misclassification later on during air pollution modeling for both measurements at samplers throughout the study area and the background setup at the monitoring station used for temporal correction of all other sites. We are unaware of any collocated PM sampling results for locations using ESCAPE methods to compare our findings. In contrast, NO_2_ and NO_x_ measurements from the continuously sampling Ogawa badges were highly correlated with collocated regulatory monitors.

For continuous Ogawa sampling, our findings are similar not only to other ESCAPE locations in Europe [24] but also to two other studies using the sampler at reference sites in El Paso, TX, another city on the U.S.–Mexico border in the western U.S., where NO_2_ concentrations were also highly correlated (r = 0.91 and 0.94) [23,37]. Interestingly, while these El Paso–based studies had concentrations more than double those of TAPS, Perth, Australia, had low pollutant concentrations and correlations nearly identical to Tucson’s [27]. Together, this indicates that Ogawa methods are accurate against reference monitors, even when sampling at relatively low concentrations in arid conditions.

Across seasons and site types in TAPS, measurements and annual mean concentrations were often less than half of the relevant National Ambient Air Quality Standards (NAAQS). Concentrations were highest in winter, much like in El Paso, TX [23,37], likely owing to thermal inversions and stagnation and a swell in anthropogenic sources from Tucson’s seasonal winter visitors [7]. Interestingly, NO_2_/NO_x_ ratios were not different by season or site type, suggesting that sources do not vary drastically over time. While street sites would be expected to have higher ratios owing to diesel vehicles, which emit a higher proportion of NO_2_ to NO_x_ compared to gasoline vehicles [38], this was only seen in the winter and intermediate seasons. In summer, increased sunlight may be increasing the conversion rate of diesel-based NO_2_ to NO [36,39]. However, the ratios are comparable to those of ESCAPE, indicating similar pollutant sources (and/or sinks), despite Europe having a larger percentage of diesel vehicles [40,41].

Similar to the case in ESCAPE, street sites generally had the highest annual mean concentrations, while rural background sites had the lowest. Differences in our study were not significant, unlike in other ESCAPE study locations [24,26,34]. Tucson’s lack of large point sources may be one explanation, combined with differences in fleet characteristics and fuel types [40,41]. In addition, Europe and Taipei are generally more densely developed, resulting in more pollution sources in a smaller area, often with street canyons reducing dilution [42,43]. For example, the City of Tucson, the most densely developed portion of the study area, has 890 people/km^2^ and no building canyons, while Oslo, Norway, has 1532 people/km^2^. Compared to Oslo, Tucson’s concentrations are slightly higher, yet they are far lower than those for cities in southern Europe such as Barcelona, Spain (37.4 μg/m^3^), with more persons at a much higher density (1,620,343 people; 16,000 people/km^2^). Relatedly, stop-and-go driving, which results in greater PM_2.5_ emissions per mile, is more common in denser areas [44,45]. Interestingly, PM means and ranges in TAPS were nearly identical to those in the coastal metro area of Perth, Australia (1.97 M persons; 307 persons/km^2^). This may be explained by Perth’s lower population density compared to Europe and the City of Tucson, as well as vehicle fleet characteristics and wind-blown dust sources [28].

It is worth noting that only 4/19 (20%) of TAPS PM sites were street sites, compared to no less than 35% in ESCAPE, likely because only certain residents had the requisite security, electricity, space, and willingness to host a PM sampler setup. Similarly, just 12/39 (30%) of TAPS NO_2_ and NO_x_ sites were street sites, versus about 50% in ESCAPE. As a result, our study may have under sampled areas that could meet PM site requirements, specifically in areas with no secure outdoor space, more density, and likely higher pollution concentrations. However, Perth, Australia, which had about 50% street sites, had pollutant concentrations nearly identical to those found in TAPS [27], suggesting more street sites would not have substantively expanded the distribution of Tucson’s concentrations.

## 5. Conclusions

In conclusion, our study illustrates how ESCAPE methods can be generally adopted and adapted to a desert metropolis with air pollution concentrations below regulatory levels with health impacts [46]. While most methods were robust to Tucson’s volatile precipitation patterns and extreme heat and aridity, study PM_2.5_ measurements were poorly correlated with those from a collocated reference monitor. This may have been because of differences in inlet heights; oversampling of high PM_2.5_ events because of non-continuous sampling; the pump operating beyond its temperature specifications; and/or other unknown variables. 

While our study has notable strengths based on the uniqueness of Tucson’s volatile climate and low-pollution airshed and the use of well-documented methods coupled with collocated regulatory monitoring, it is not without weaknesses. Our pumps were unable to produce diagnostic output that might have confirmed our suspicions of mechanical issues resulting from high ambient temperatures, potentially leading to poor correlations between collocated study and regulatory monitors. Meteorology data obtained geographically closer to the regulatory monitor could have shed more light on factors that may have influenced poor correlation for PM_2.5_ with less potential for highly spatially variable weather patterns. PM_10_ concentrations from a regulatory monitor were also unavailable at the regulatory monitor site, blunting our ability to investigate the relationships between study and regulatory setups. 

As climate change makes meteorological patterns more extreme, much like Tucson’s is today [10,11,12,13], it will be critical to reevaluate air pollution monitoring methods. Future studies should take advantage of collocating with regulatory monitor setups to help ensure that study samplers are field validated against “gold standard” methods, especially those meant to more accurately measure low concentrations. This is even more important for low-pollution airsheds, where there is less room for error. Without this assurance, sampling bias in different meteorological or pollution scenarios may compound in later air pollution and health effects models. Low air pollution concentrations with intra-city spatial precision are rare [47], yet they are critical in assessing the impacts of reductions in air pollution.

## Figures and Tables

**Figure 1 ijerph-19-03173-f001:**
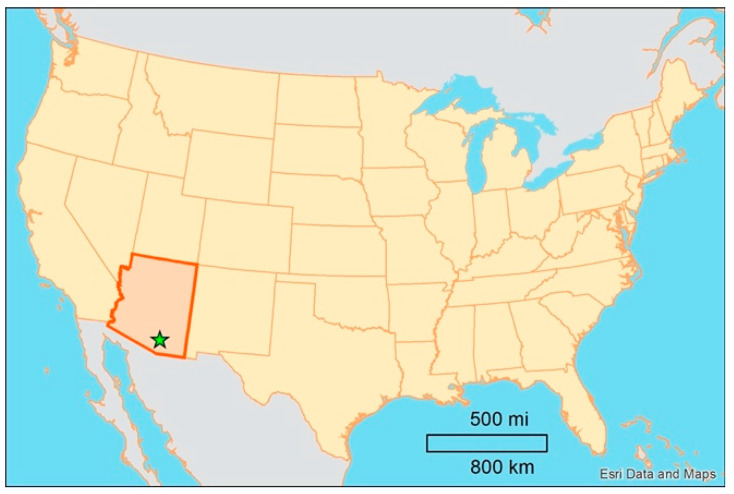
The Tucson metropolitan area (starred) in the bolded state of Arizona (AZ).

**Figure 2 ijerph-19-03173-f002:**
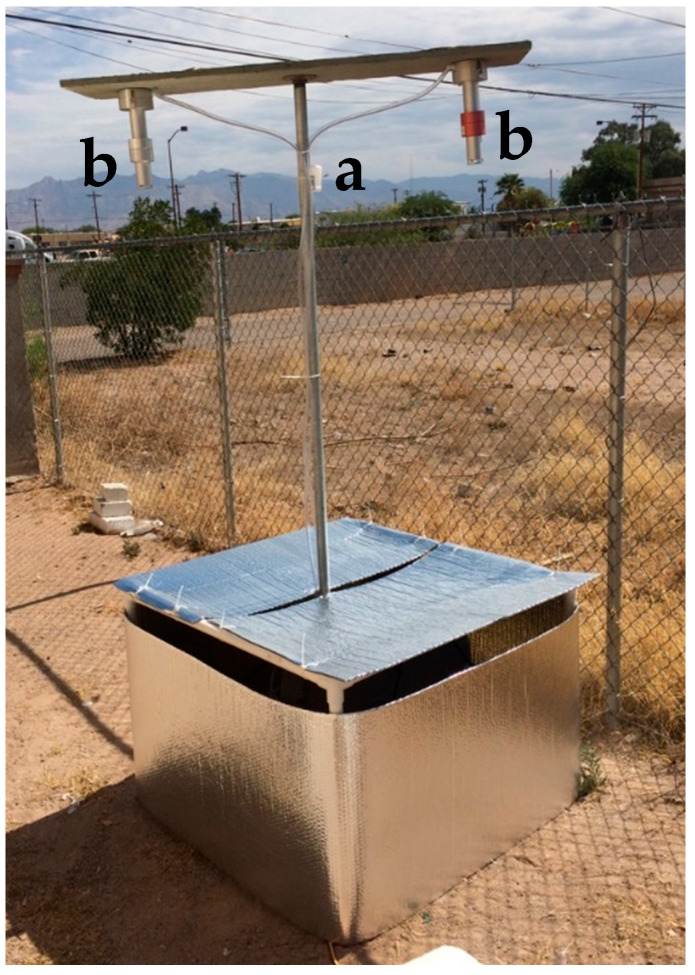
A typical sampler setup for NO_2_, NO_x_, PM_2.5_, and PM_10_. The Ogawa Badge Sampler (a) is attached near the top of the central post, while Harvard impactors hang, inlet down, from each end of the horizontal plank (b). Leland Legacy pumps are in a pump case concealed by UV-reflective foil insulation.

**Figure 3 ijerph-19-03173-f003:**
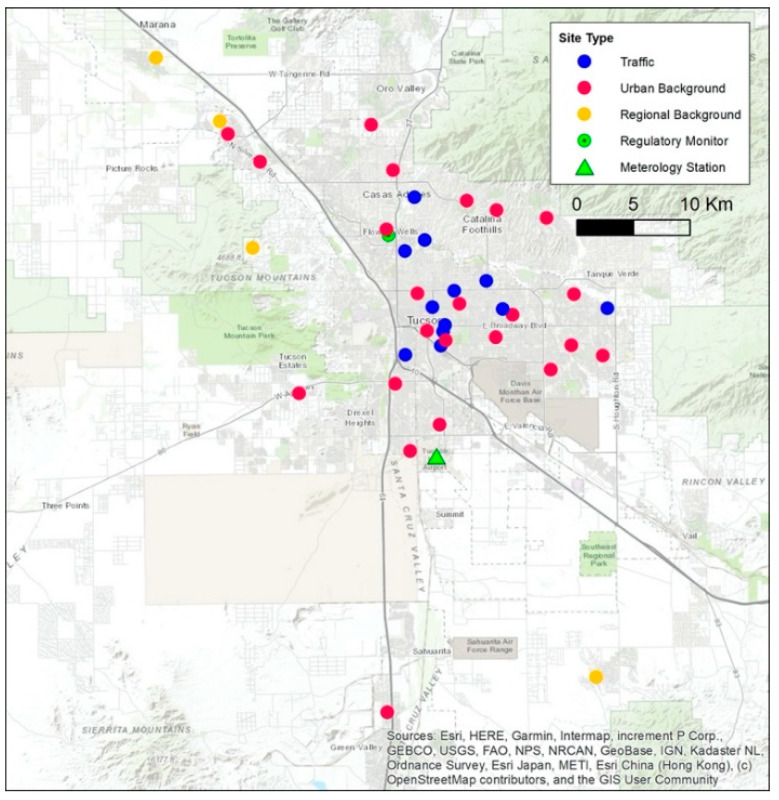
Monitoring sites in the Tucson Air Pollution Study area, including the regulatory monitoring site and meteorology station.

**Figure 4 ijerph-19-03173-f004:**
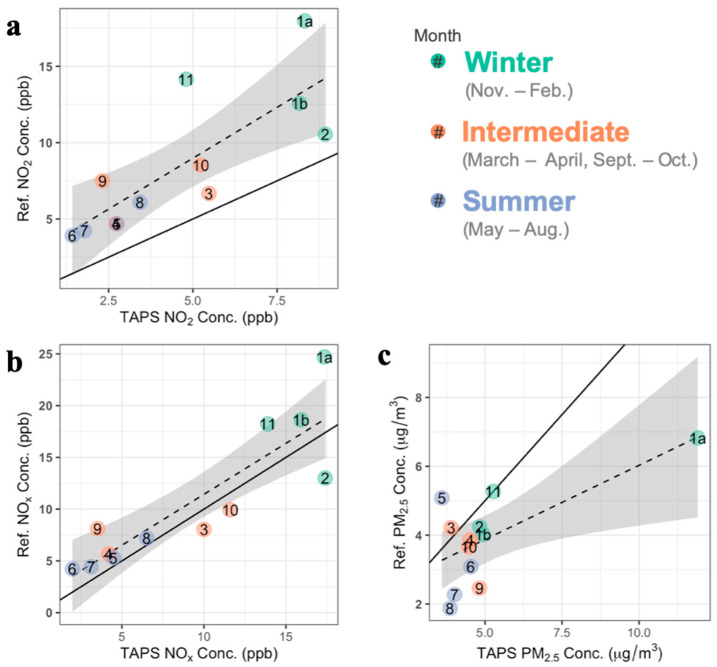
Comparison of measured concentrations of (**a**) NO_2_, (**b**) NO_x_, and (**c**) PM_2.5_ by month for the collocated Tucson Air Pollution Study (TAPS) and reference (Ref.) monitors. Points are labeled with month. The solid line is the identity line; the dashed line, the linear regression line; and the gray shaded area, the 95% confidence interval. Months 1a and 1b are the first and second two-week periods in January, respectively.

**Figure 5 ijerph-19-03173-f005:**
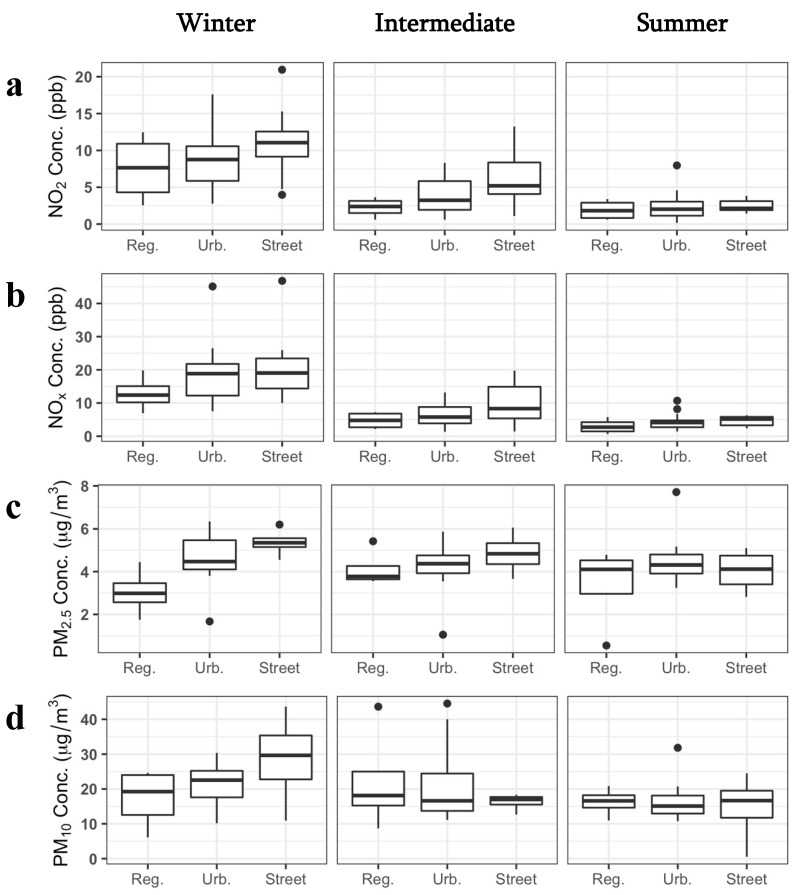
Pollutant measures by season and site type for (**a**) NO_2_, (**b**) NO_x_, (**c**) PM_2.5_, and (**d**) PM_10_. In boxplots, the bold horizontal line represents the median, lower and upper ends of the box represent the first and third quartiles, the upper whisker extends to 1.5 × IQR above the box,. and the lower whisker extends to 1.5 × IQR below the box. Reg.: regional background; Urb.: urban background.

**Figure 6 ijerph-19-03173-f006:**
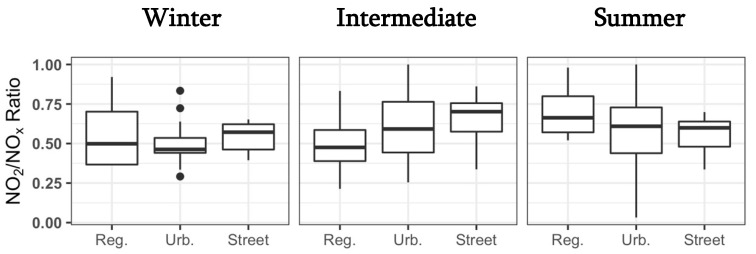
NO_2_/NO_x_ ratios by season and site type. In boxplots: the bold horizontal line is the median, lower and upper ends of the box are the first and third quartiles, the upper whisker extends to 1.5 × IQR above the box, and the lower whisker extends to 1.5 × IQR below the box. Reg.: regional background; Urb.: urban background.

**Table 1 ijerph-19-03173-t001:** Hourly meteorological measures by sampling period month.

	Precip. (mm)		Sea-Level Pressure (hPa)	Dew Point (°C)	Temp. (°C)	Wind Speed (m/s)	Visibility (km)
Sampling Month	Total	Mean (SD)	Mean (SD)	Mean (SD)	Mean (SD)	Mean (SD)	Mean (SD)
11	3.30	0.009 (0.09)	1016 (3.81)	−1.65 (4.60)	14.6 (6.40)	3.42 (2.09)	16.1 (0.41)
1a	1.02	0.002 (0.04)	1020 (4.00)	−2.22 (3.60)	9.95 (5.81)	2.60 (1.64)	16.1 (0.28)
1b	4.57	0.01 (0.16)	1020 (5.76)	−7.51 (3.72)	13.5 (8.00)	3.40 (2.28)	16.1 (0.30)
2	0	-	1015 (3.30)	−6.70 (2.64)	18.5 (6.55)	3.20 (1.74)	16.1 (0)
3	6.45	0.02 (0.22)	1013 (4.17)	−5.18 (6.05)	19.1 (6.50)	3.23 (1.95)	16.1 (0.35)
4	0	-	1009 (4.21)	−5.05 (4.17)	21.3 (5.83)	3.89 (2.41)	16.1 (0)
5	0	-	1008 (3.65)	−4.60 (3.42)	28.0 (7.06)	3.25 (1.90)	16.1 (0)
6	56.1	0.12 (1.00)	1009 (2.96)	14.0 (2.89)	31.9 (5.00)	3.24 (1.51)	16.0 (0.80)
7	80.6	0.20 (1.25)	1009 (3.42)	19.8 (0.52)	29.8 (4.76)	3.10 (1.78)	15.9 (1.04)
8	7.97	0.02 (0.16)	1012 (3.02)	16.3 (0.81)	25.6 (5.00)	3.49 (2.19)	16.1 (0.27)
9	38.4	0.11 (0.92)	1009 (2.53)	12.0 (4.30)	27.8 (4.95)	3.26 (2.01)	15.9 (1.44)
10	40.4	0.11 (0.74)	1011 (3.26)	9.68 (3.36)	20.4 (5.01)	3.26 (2.18)	15.9 (1.00)

SD: standard deviation; months 1a and 1b are the first and second two-week periods in January, respectively.

**Table 2 ijerph-19-03173-t002:** Summary of measured and temporally corrected annual mean pollutant concentrations (ppb) by study site type.

			NO_2_		NO_x_			PM_2.5_		PM_10_	
	Site Type	n	GM (GSD)	Range	GM (GSD)	Range	n	GM (GSD)	Range	GM (GSD)	Range
Measured	Reg.	4	3.42 (1.92)	1.35–6.28	6.32 (1.60)	3.84–10.3	4	3.38 (1.41)	2.06–4.45	18.3 (1.24)	13.6–21.9
	Urb.	23	4.57 (1.53)	1.98–11.3	8.78 (1.48)	3.89–21.9	11	4.34 (1.24)	3.23–6.73	17.1 (1.61)	5.35–30.1
	Street	12	6.22 (1.58)	2.33–10.7	11.0 (1.56)	4.79–23.0	4	4.73 (1.10)	4.20–5.18	18.5 (1.57)	9.65–26.6
Corrected	Reg.	4	4.09 (2.05)	1.61–7.51	8.00 (1.50)	4.64–11.8	4	3.27 (1.21)	2.67–4.17	19.8 (1.19)	16.2–24.0
	Urb.	23	5.11 (1.79)	1.36–13.0	10.7 (1.79)	1.56–30.2	11	3.92 (1.28)	2.52–5.72	18.8 (1.57)	7.25–34.8
	Street	12	7.39 (1.72)	2.80–15.8	14.3 (1.65)	6.38–33.6	4	4.83 (1.19)	4.18–6.19	21.0 (1.55)	11.1–29.1

GM: geometric mean; GSD: geometric standard deviation.

## Data Availability

Data are available upon reasonable request by contacting N.L. at lothrop@arizona.edu.

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
