# Peer review of "Sampling Low Air Pollution Concentrations at a Neighborhood Scale in a Desert U.S. Metropolis with Volatile Weather Patterns"

_ijerph, 2022, doi:10.3390/ijerph19063173_

Round 1

Reviewer 1 Report

This paper explores exposure assessment for a low air pollution airsheds with extreme weather events. Overall, the topic of this paper is helpful and the paper could contribute to current discussions on exposure assessment for different contexts. The manuscript is well-written, and the reviewer has the following comments that may help improve the manuscript.

  1. Introduction
  • It would be possible to add some discussions on why knowledge in low-pollution airsheds is also important for exposure assessment and epidemiology.
  • Identifying some examples of sampling methods would help offer a context of exposure assessment.
  1. Materials and Methods
  • It might be helpful to specify the recommended sites from ESCAPE study and highlight the rationale behind it. For example, readers might ask why 40 sites is enough.
  • Why industrial sites are not included? Does it mean that there are not so many industrial sites OR it is to best match the locations of the cohorts?
  • It might be helpful to include some descriptions of the pollutants. For example, what are the potential emission sources and what is the general concentration levels in the US or Western US? This will give readers a context of this unique arid environment.
  • Figure 2 can be improved to specify the sampler instrument for those who are not familiar with the sampling field.
  1. Results
  • Since this is a field measurement study, adding a map of samplers would help readers understand the spatial coverage.
  • The authors may want to clarify why the Total of month 2, 4, 5 are 0.
  1. Discussion
  • It would be interesting to add some discussions on recommendations for future sampling campaigns in similar areas and policies for model development based on these sampling measurements.
  • How does the low-pollution context differ from the general context? What are the potential implications for exposure assessment and epidemiological studies?

Reviewer 2 Report

The work is well handled and structured by the authors. There focus of the study is useful to publish but a few edits are suggested to help with the flow of the paper. 

The correlation coefficient is represented by r, not R2. These errors need to be fixed.

Is the R2 value written in the abstract of the study a correlation value or a statistical analysis value? R-squared (R2) is a statistical measure that represents the proportion of variance for an input factor or an output factor explained by factors (variables) in a regression model. R (r), the correlation between the input (independent, predictor) variable (factor), x, and the response (output, dependent) variable (factor), y.

The introduction part of the study can be detailed.

The conclusion part of the study should be expanded. The conclusion of the paper should improve as to what are the limitations and shortcomings of this work and in what areas is further work needed.

Reviewer 3 Report

While your manuscript has apparently undergone some sort of editorial screening and considered to be of sufficient quality to be sent out for review.
While the topic is good and the analysis is well-done, the importance of the research needs to be better explained.

Specific comments:

The study involved a valuable and an important number of data and variables. I think that such dataset is not fully analyzed. I recommend applying statistical tools, particularly multivariate techniques, to solve this complex case of multiple pollution sources. Multivariate techniques include among others: principal components analysis (PCA), cluster analysis (CA), and canonical correlation analysis (CCA).

Line 223. "NO2/NOx ratios were not significantly different by site type or season", and if NO2/NOx ratio were significant, what would that mean. This also applies to PM2.5/PM10 ratio. If these ratios are greater than 1, what will it mean?

why don't the authors study the diurnal variation of pollutants?

Reviewer 4 Report

This paper is interesting and should be published after a few issues are addressed.

 - A map of sampler locations should be shown.  It should be easy for the reader to determine the number, geographical distribution, and attributes of the samplers. 

 - Please justify why  geometric means were used for pollutant concentrations rather than arithmetic means.

Table 1 (met measurements)
 - Pressure is typically presented in hPa units, rather than kPa
 - The pressures presented appears to be station pressure, which provides little insight on weather conditions from a meteorological perspective.  Meteorological convention is to present sea-level pressure.
 - RH (relative humidity) is a strong function of both temperature and moisture.  On a day with little change in the absolute moisture (water vapor) concentration, RH can change drastically.  It's not possible to understand/interpret the mean RH without undestanding the details of its calculation.  

 - Figure 3: please define "TAPS"

 - It's difficult to interpret Figures 4 and 5, and Table 2, without seeing a map of station locations.

 - Since monitor inlet height was described, it would be useful to include a table, perhaps in Supplemental Information, of relevant details (including inlet height) of each sampling location.

 - There was discussion of pumps possibly operating beyond their temperature specifications.  Can any non-speculative information be presented?  Can you present hourly temperature observations during periods for which this temperature exceedance was expected to have occurred?  Are there any instrument diagnostics that can shed light on this?

Round 2

Reviewer 4 Report

The authors have addressed all of my concerns, and I now recommend that the paper be published.  

Author Response

Thank you for your feedback and help improving the manuscript.